



# Evaluation of a Partector Pro for atmospheric size distribution and number concentration measurements at an urban background site

Christof Asbach[1], Ana Maria Todea[1], Heinz Kaminski[1]

[1] Filtration & Aerosol Research Department, Institut für Umwelt & Energie, Technik & Analytik e. V. (IUTA), 47229 Duisburg, Germany

*Correspondence to*: Christof Asbach (asbach@iuta.de)

**Abstract.** Number size distributions, total number concentrations and mean particle sizes have been measured for 70 days at an urban background site in Mülheim-Styrum, Germany with a handheld Partector Pro and a TSI mobility particle size spectrometer (MPSS). The aim of the study was to evaluate the performance of the Partecor Pro against the MPSS. The results show that the size distributions, measured with the Partector Pro agree with the MPSS mostly within ±25% for particle sizes below 113.5 nm, whereas higher, systematic differences were observed for larger particles. The measurement accuracy was shown to be dependent on the geometric mean diameter and the geometric standard deviation of the aerosol. Best results were found for the most abundant size distributions with geometric mean particle diameters ≥30 nm and geometric standard deviations larger than 1.8. The total number concentration, measured by the Partector Pro was found to be in excellent agreement with the MPSS with a slope of the linear fit of 0.9977 and a regression coefficient of $R^2 = 0.9956$. The agreement of the geometric mean particle diameter, determined with the Partector Pro and the MPSS was good, but moderately dependent on the particle size distribution. For mean particle sizes between 20 nm and 50 nm, the bias was within ±15%. Higher deviations of up to 30% were observed when the geometric mean particle sizes exceeded 70 nm and when the geometric standard deviations exceeded approximately 2.7.

## 1 Introduction

Exposure to air pollution in general and particulate matter (PM) in particular has been a major health concern for many decades. The Lancet commission reported in 2018 that air pollution causes 6.5 million premature deaths worldwide every year (Landrigan, et al., 2018) and ranked it as fifth leading cause of death (GBD 2013 Risk Factor Collaborators, 2015). A multitude of epidemiological and toxicological studies have shown clear correlations between increased PM concentrations and adverse health effects (Dockery, et al., 1993; Dockery, Health Effects of Particulate Matter, 2009; Pope III & Dockery, 2006; Rückerl, Schneider, Breitner, Cyrys, & Peters, 2011; Lelieveld, Evans, Fnais, Giannadaki, & Pozzer, 2015). Legislation currently only requires monitoring of mass concentrations of the particulate matter fractions $PM_{10}$ and $PM_{2.5}$, i.e. particles with an aerodynamic diameter <10 µm and <2.5 µm, respectively. However, in recent years, there has been increasing evidence that mass concentrations are insufficient predictors of the health effects of exposure to airborne particles. The relationship between



particle size and potential health effects remains incompletely understood and sometimes generates controversy. For example, while Iskandar et al. (Iskandar, et al., 2012) concluded that asthma-related hospital admissions in children were correlated with concentrations of coarse and fine particulate matter but not ultrafine particles (UFP, particles <0.1 µm), Franck et al. (Franck, Odeh, Wiedensohler, Wehner, & Herbarth, 2011) postulated that the smaller the particles, the stronger the effect on

cardiovascular disease in general.

UFP in the urban atmosphere originate predominantly directly from combustion processes, e.g., road traffic (Rivas, et al., 2020) (Kumar, et al., 2014), but can also arise naturally from nucleation processes in outdoor air (Kulmala, 2003). In the recent years, UFP emissions from aircraft have received increased scientific and public attention (Stacey, 2019; Hudda & Fruin, 2016).

To account for the new body of knowledge regarding the health effects of particularly smaller particles, the additional measurement of ultrafine particles has been recommended (Birmili, et al., 2014; Peters, Wichmann, Tuch, Heinrich, & Heyder, 1997). Whereas larger particles dominate the commonly measured PM mass concentrations, UFP in the atmosphere typically appear in high number, but low mass concentrations. UFP concentrations in the atmosphere are therefore measured in terms of the total particle number concentration or number size distribution. As of now, no regulatory limit values for UFP

concentrations exist, among others due to the lack of sufficient data for epidemiological studies, stemming from the lack of measurement obligation for UFP. In 2021, the World Health Organization (WHO) called for UFP measurements to be taken and requested the use of standardized measurement methods. In 2022, the European Union consequently amended its air quality directive 2008/50/EG and now requests at least one UFP measurement site per 5 million inhabitants at locations with expected high concentrations and at supersites. The European standardization organization CEN has issued two Technical

Specifications with the aim of harmonizing the sampling and measurement techniques used for measuring the total number concentration and number size distribution of ambient UFP using condensation particle counters (CPC) (CEN/TS 16976:2016, or draft standard prEN 16976:2023) (DIN, 2023) and mobility particle size spectrometers (MPSS) (CEN/TS 17434:2020) (CEN, 2020).

Besides potential future obligatory UFP measurements, the knowledge on UFP number concentrations and size distributions

is of high interest in many applications, such as mobile or screening measurements. MPSS systems cannot be used for such studies due to their large size and weight and the requirement for mains power supply. In addition, the CPC requires regular refilling of the working fluid reservoir. MPSS systems are furthermore very cost-intensive and may therefore not be affordable in each case. An attempt to lower the cost and maintenance of ambient UFP measurements was developed in the UFIPOLNET project (Hillemann, 2013). This UFP-monitor measured UFP size distribution in six size bins from 20 nm to 500 nm and used

a unipolar diffusion charger to bring the particles to a defined charge level, a differential electrical mobility classifier (DEMC) to classify the particles based on their electrical mobility and an aerosol electrometer to determine the concentration of the classified particles. This device had been commercialized by TSI as model 3031, but has been discontinued in the meantime. The recently introduced Partector Pro (naneos GmbH, Windisch, Switzerland) is a small and lightweight instrument (142 × 88 × 34 mm³, 415 g), that can be battery operated and determines the number concentration, lung deposited surface



area (LDSA) concentration, geometric mean particle size and number size distribution of airborne particles in a nominal size range between 10 nm and 300 nm. However, no size-selective inlet is used and thus larger particles may enter the device and interfere with the measurement. The number size distribution is delivered in eight size bins. Unlike an MPSS, the Partector Pro does not require a radioactive or soft X-ray charger and no working fluid. It only requires about 0.5 W electrical power and may thus be operated for a long time, independent of a mains power supply, if powered by a solar panel.

In the study, presented here, a Partector Pro was operated continuously for around 70 days from 22$^{nd}$ March to 1$^{st}$ June 2023 at an urban background site in Mülheim-Styrum in Germany. An MPSS system, fully compliant with CEN/TS 17434:2020, measured number size distributions in a size range from 10 nm to 800 nm alongside and was used as a reference instrument to evaluate the performance of the Partector Pro in terms of number concentration, mean particle size and number size distribution. The performance of the Partector Pro is evaluated by comparing it to criteria set in the CEN/TS 17343 for

measuring the number size distribution and CEN/TS 16976:2016/prEN16976:2023 for the total number concentration. The Partector Pro is not intended to replace MPSS and CPC systems and does not fulfil their design criteria and several performance criteria according to the aforementioned standards. Nevertheless, these were considered here to evaluate the Partector Pro, as they are the only ones that provide dedicated criteria for comparable devices for atmospheric particle measurements. In comparison to MPSS and CPC, the main advantages of the Partector Pro are its small size, low price, low power requirement

and lack of need for a working fluid and regular maintenance, which however require compromises in accuracy and size resolution. The Partector Pro is thus mainly intended for mobile and indicative measurements rather than the control of potential future limit values.

## 2 Measurement location

The measurement container is located in Mülheim-Styrum in the western Ruhr area in Germany (Coordinates ETRS89

51.453459° North, 6.86505° East), directly adjacent to an official measurement station (Code DENW038, short name STYR) of the environmental protection agency of the federal state of North Rhine Westphalia (Landesamt für Natur, Umwelt und Verbraucherschutz, LANUV) (LANUV, 2018). The district of Styrum is located at the northern edge of the city of Mülheim an der Ruhr. Within a radius of approx. 12 km around the measurement location, the city areas of Mülheim an der Ruhr (south, approx. 171,000 inhabitants), Duisburg (west, approx. 500,000 inhabitants), Oberhausen (north, approx. 211,000 inhabitants),

Bottrop (northeast, approx. 117,000 inhabitants) and Essen (east, approx. 583,000 inhabitants) are almost completely covered. In the direct vicinity of the measuring station, in a northerly direction at a distance of approx. 50 m, there is a youth sports and leisure field; in all other directions, the site is surrounded by residential areas. Approximately 220 m north of the measuring station runs the busy A40 freeway in west-east direction with approx. 120,000 motor vehicles and 8,000 heavy duty vehicles daily (Ministerium für Verkehr des Landes Nordrhein-Westfalen, 2022). The junction "Mülheim-Styrum" is at a distance of

approx. 430 m in a north-westerly direction and connects the A40 with the B223 federal highway, which runs in a north-south direction. Approx. 600 m to the east and 750 m to the southeast are large steel processing plants (see Figure 1).





**Figure 1**

Potential other sources at a larger distance from the station include the international airport in Düsseldorf approximately 20 km south, two steel mills in Duisburg approximately 14 km southwest and 10 km west, respectively, Europe's largest inland harbor in Duisburg 7.5 km west and a coking plant in Bottrop approximately 10 km north of the measurement location (Asbach, Kuhlbusch, Quass, & Kaminski, 2020).

The measurement station is classified as an urban background site (Lenschow, et al., 2001) according to the EU Commission Decision 2011/850/EU and is part of the German Ultrafine Aerosol Network (GUAN) (Birmili, et al., 2009). UFP measurements have been conducted continuously at this station since 2009 (Asbach, Kuhlbusch, Quass, & Kaminski, 2020). The highways A40 in the north and northeast and B223 in the west were identified as main sources of UFP. Due to the location in a densely populated region and with various potential particle sources in the vicinity, the UFP concentrations were repeatedly

the highest among all GUAN urban background sites and at a similar level like those measured at roadside locations (Sun, et al., 2019).

During the measurement period, the wind direction was most often from north-east (NE) and east-north-east (ENE) at low to moderate wind speeds. The highway A40 was thus likely the main UFP source during periods of this wind direction. Temperature, relative humidity and precipitation during the measurement period are presented in Figure S 1 and the wind rose

in Figure S 2 in the supplementary material.

## 3 Methods

### 3.1 Sampling system

The measurement station is equipped with a UFP sampling system (TSI model 3750200) that is compliant with CEN TS 17343:2020 and consists of a USEPA $PM_{10}$ inlet, followed by a $PM_{2.5}$ sharp cut cyclone. Both are operated at a flow rate of

1 m³/h (16.67 l/min) and mounted above the container roof at a level of approximately 4 m above ground. Of the total flow, 4 l/min are passed through a single tube Nafion® dryer, which according to the manufacturer assures the relative humidity to be below 40% at its outlet. The dried aerosol flow can be split into four individual flows in a 4-way flow splitter (TSI model 3708). One exit of the flow splitter is connected to the Mobility Particle Size Spectrometer (MPSS, TSI model 3938W50-CEN) which draws 1 l/min and another exit to the Partector Pro, which has a total inlet flow rate of 0.5 l/min. The other two

exits of the flow splitter are combined to draw an excess flow of 2.5 l/min in order to maintain the total flow rate of 4 l/min through the dryer.



Particle losses in the inlet system have been determined by the manufacturer and a particle loss correction is included in the MPSS software (AIM11SMPSMONITOR). In order to use the same correction for the size distributions measured by the Partector Pro, a measured number size distribution has been exported with the sampling system loss correction enabled and
disabled, respectively. The particle size dependent losses were calculated and fitted to an exponential function using Origin Pro 2022, resulting in the following empirical fit equation for the losses $l(d_p)$ in the sampling system (see Figure 2):

$$l(d_p) = 0.65103 \cdot exp\left(-\frac{d_p/nm}{5,08574}\right) + 0.16303 \cdot exp\left(-\frac{d_p/nm}{18,76118}\right) + 0.02658 \cdot exp\left(-\frac{d_p/nm}{104.97138}\right) + 0.00247 \qquad (1)$$

The correlation coefficient $R^2$ of the fit is 0.99997, showing that the equation provides an accurate estimation of the losses.

**Figure 2**

### 3.2 Mobility Particle Size Spectrometer

The particle number size distributions were measured in a size range from 10 nm to 800 nm (electrical mobility diameter) with the MPSS. An MPSS classifies particles based on their electrical mobility by exposing them to an electrical field in a differential electrical mobility classifier (DEMC, also known as differential mobility analyser DMA (Liu & Pui, 1974) and determines the number concentration of the mobility-classified particles with a condensation particle counter (CPC) (Agarwal, 1980; McMurry, 2000). By ramping the electrical field strength in the DEMC, the electrical mobility range corresponding to
the abovementioned particle size range is covered. A data deconvolution algorithm is used to correct for multiply charged particles and thus determine the number size distribution from the mobility distribution (Hoppel, 1978) (Fissan, Helsper, & Thielen, 1983). The type of MPSS used here is also known as scanning mobility particle sizer (SMPS) (Wang & Flagan, 1990). The MPSS and the sampling system used are fully compliant with the requirements of CEN/TS 17343:2020 (CEN, 2020) for the determination of the particle number size distribution of atmospheric aerosol with an MPSS. It consists mainly of an [85]Kr
neutralizer (initial activity of 370 MBeq; TSI model 3077A), a differential electrical mobility classifier (DEMC, TSI model 3083) and a full flow condensation particle counter (CPC, TSI model 3750, $d_{50}$ = 10 nm). The MPSS and CPC have been calibrated at the World Calibration Center for Aerosol Physics (WCCAP) in Leipzig, Germany in December 2022 and their uncertainties found to be in full agreement with the requirements set in CEN/TS 17343. The MPSS was thus used as a quasi-reference for the evaluation of the performance of the Partector Pro. The aerosol flow rate of the DEMC was set to 1 l/min and
the sheath flow rate to 4.8 l/min, resulting in a covered particle size range from 10.0 nm to 805.8 nm (lower and upper limit of the smallest and largest size bin, respectively). The midpoints of the size bins are from 10.18 nm to 791.48 nm. The scan time was set to 240 s and a new measurement started every 5 minutes. No impactor was used at the device inlet. The MPSS was operated with the manufacturer's software (AIM11SMPSMONITOR) that controlled the device functions, recorded the data



and automatically exported them to a connected external laptop every day at midnight. The software was set to correct for
diffusional particle losses in the sampling system (see above) and inside the device and to apply the multiple charge correction
algorithm prior to exporting the data. Size distributions were recorded and exported with a size resolution of 64 bins per size
decade.

### 3.3 Partector Pro

The Partector Pro (naneos GmbH, Windisch, Switzerland) is a further development of Partector and Partector 2. All Partector
models are small and portable devices that can be battery-operated. They use unipolar corona diffusion chargers to charge the
aerosol particles to a known, particle size dependent charge distribution. The charger is operated in a chopped mode, i.e. it is
switched on and off in short intervals to produce charged particle parcels. As these charged parcels enter and leave an induction
tube, they induce a negative and positive voltage peak, respectively. The heights of these peaks are directly proportional to the
total charge concentration, which is determined as the half peak-to-peak value (Fierz, Meier, Steigmeier, & Burtscher, 2014).
For particle sizes between approximately 20 nm and 400 nm (Asbach, Fissan, Stahlmecke, Kuhlbusch, & Pui, 2009), the charge
concentration and consequently the total current produced by the charged particles is directly proportional to the lung deposited
surface area (LDSA) concentration of the particles (Fissan, Neumann, Trampe, Pui, & Shin, 2007), which can be measured
with around ±30% uncertainty (Todea, Beckmann, Kaminski, & Asbach, 2015) (Todea, et al., 2017). The LDSA concentration
is thus the primary measurand of all three Partector models. Partector 2 uses the same overall principle, but with a dual stage
charge measurement. Downstream of the first induction tube, the aerosol passes an aerosol manipulator, which essentially is
an electrostatic precipitator and exposes the particles to a constant electric field to remove a certain fraction of charged
particles. The current stemming from the remaining aerosol fraction is measured in a second induction tube. The ratio of the
two measured currents is dependent on the mean particle size. Based on the measurement of the two currents and assumptions
on the particle size distribution, the mean particle size and the particle number concentration are determined by the Partector
2 in addition to the LDSA concentration, similar to the measurement with other devices with either mechanical (Fierz, Houle,
Steigmeier, & Burtscher, 2011) or electrical aerosol manipulator (Marra, Voetz, Kiesling, & H.J., 2010). Unlike in the Partector
2, the Partector Pro does not apply a constant electric field strength in the manipulator, but switches between four different
field strengths (voltages) in order to remove different size fractions from the aerosol and consequently determines the number
size distribution of the aerosol. A similar concept has been used in the past by the Electrical Aerosol Analyzer (Liu & Pui,
1975; Qi, Chen, & Greenberg, 2008). The current integration time for each field strength can be 2 s, 4 s, 8 s, 16 s or 32 s. In
addition, the instrument waits 2 s after a voltage change before the current measurement for the next voltage setting is started.
The size distribution is provided in eight logarithmically spaced size bins with midpoints between 10 nm and 300 nm. The
data for a new size distribution is provided whenever the current integration time for a voltage setting in the manipulator has
passed, i.e. every 4 s, 6 s, 10 s, 18 s or 34 s, respectively. Geometric mean particle diameter and total number concentration
are determined from the measured size distribution. All measured data are recorded and stored to an SD card.



The Partector Pro used during the measurements in Mülheim-Styrum was originally a Partector 2 (S/N 8031) that has received a hardware (Version 3.1) and firmware (Version 275.9) upgrade by the manufacturer. The device thus belongs to the very first generation of the Partector Pro. It was operated by means of an external USB power supply and using its internal pump. The integration time was set to 4 s and consequently, a "new" size distribution was recorded every 6s.

**4 Results and discussion**

The MPSS and Partector Pro clocks were synchronized prior to the measurements. After the 70 days of measurements, the clock of the Partector Pro was found to go approximately 5 minutes and 30 s ahead. The time stamps of the Partector Pro files were corrected accordingly. All MPSS and Partector Pro data were initially averaged on an hourly basis. The size distributions measured with the Partector Pro were corrected for the sampling inlet losses, using equation (1). The CPC of the MPSS ran

out of Butanol during the time from 11th May 06:00 to 14th May 16:00. Consequently, these data were excluded from further analyses.

**4.1 Number size distribution**

Figure 3 shows contour plots of the hourly averages of the size distributions measured with the MPSS and Partector Pro. For this representation, the size distributions measured with the MPSS have been limited to the size range from 10 nm to 300 nm

in order to match the size range of the Partector Pro. The contour plots for both devices show very similar patterns, although the contours of the MPSS data appear sharper than the ones of the Partector Pro. This is caused by the different size resolutions of the two devices, i.e. 95 channels (MPSS) vs. 8 channels (Partector Pro) in the displayed size range. The highest concentrations are most of the time observed in the size range below 50 nm, which is in good agreement with the results of the long-term measurements at this station (Asbach, Kuhlbusch, Quass, & Kaminski, 2020) and mainly caused by the nearby

traffic sources. Only in few cases, the size distribution is shifted towards larger particle sizes. A prominent example can be seen in Figure 3 on 29th April around noon, which also coincides with a high total number concentration. This singular event was likely caused by observed barbecue activities on the adjacent sports field.

**Figure 3**

The average number size distribution during the entire measurement period measured with the MPSS and Partector Pro are shown in Figure 4 in each device's original size range and resolution. The size distribution measured with the MPSS is in general good agreement with the results at this measurement location in previous years. The decrease of the concentration towards the smallest sizes shows that possible nucleation events did not have a noticeable effect on the average size





distribution, unlike e.g. in April and May 2018, when nucleation had a strong impact on the monthly average size distributions (Asbach, Kuhlbusch, Quass, & Kaminski, 2020).

For ambient UFP measurements with an MPSS, CEN/TS 17343:2020 allows a deviation of up to 50% for particle sizes between 10 nm and 20 nm, up to 10% for sizes between 20 nm and 200 nm and up to 20% for sizes between 200 nm and 800 nm. This uncertainty range shall be determined by averaging the number size distributions over a time span of at least 8 hours and is

225 marked with grey shade in Figure 4. It can be seen that the average concentrations in all Partector Pro size bins with midpoints below 113.5 nm are within the specified uncertainty range of the CEN/TS, considering that the relatively freshly calibrated MPSS serves as a quasi-reference. However, the concentrations in these size bins are below the reference concentrations. In contrast, the concentrations in the larger size bins exceed the max. allowed uncertainty.

**Figure 4**

For a clearer graphical representation of the agreement of the two devices, Figure 5 shows the bias of the average concentrations per Partector Pro size bin along with the standard deviation as error bars for the hourly averages (black) and the daily averages (red). Daily averages have been added to account for the minimum duration of 8 h for comparison measurements according to

235 CEN/TS 17343:2020, although the differences to the hourly averages are rather small. The grey shaded area represents the allowed uncertainty range according to the CEN/TS. For the calculation of the bias, the concentrations, measured with the MPSS in the corresponding size bins have been summed up. Table 1 provides an overview of the eight Partector Pro size bins along with their lower and upper limits and the corresponding MPSS size bins that have been summed up for the calculation of the bias.

**Table 1, Figure 5**

Figure 5 shows that while the averages of the hourly and daily concentrations measured with the Partector Pro for particle sizes below 100 nm are all within the specifications of CEN/TS 17343:2020. The standard deviations, shown as error bars, for the

245 size bins between 20 nm and 100 nm, however, in most cases exceed the stricter requirement for a deviation below 10%, but are mostly within ±25%. The deviation of the concentrations in the smallest size bin (10 nm) is higher and on average –38.5% for the hourly and -43.4% for the daily averages, respectively. It should be kept in mind that the CEN/TS is intended for sophisticated MPSS systems and not simplified devices like the Partector Pro, which requires compromises to be made. Consequently, the results for sub-100 nm particles can be considered to be in reasonably good agreement with the reference.

However, the bias for larger particles is higher and with larger standard deviations. This bias appears systematic such that concentrations in the 113.5 nm size bins are almost always overestimated, on average by +13.1% for the daily average and thus just slightly higher than allowed according to the Technical Specification. In contrast, the concentrations in the 188.6 nm and 300 nm size bins are rather strongly underestimated, on average by -57.7% and -23.6%, respectively. The number



concentrations in the two largest size bins should thus be interpreted with caution. It can only be speculated about the possible

reasons for this behaviour, since the data deconvolution algorithm used by the Partector Pro is undisclosed. A possible reason may be the decreasing particle size dependence of the electrical mobility with increasing particle size of particles charged by a unipolar diffusion charger (Levin, et al., 2015). Another possible reason may be the broad charge distributions of particles downstream of a unipolar (Kaminski, et al., 2012) compared with bipolar charger (Fuchs, 1963; Wiedensohler, 1988), which increases uncertainties in the multiple charge correction (Hoppel, 1978). The Partector Pro does not apply a systematic multiple

charge correction to its four mobility bins, like an MPSS (according to personal communication with Martin Fierz, naneos GmbH), but uses an empirical approach for the correction. Due to the rather broad width of the mobility channels and the charge distribution, the outcome of this correction is much more dependent on the size distribution of the aerosol than in MPSS systems. It should further be noted that the Partector Pro used here was of the very first generation. Figure S 3 in the supplementary material shows the bias of the concentrations measured in each of the eight Partector size bins as a function of

the geometric mean diameter, Figure S 4 as a function of the geometric standard deviation. The bias of the 16.3 nm size bin shows the lowest dependence on the mean particle size. The bias in the 10 nm size bin is highest for mean particle diameters around 30 nm, whereas the bias in all other size bins, except for the 184.6 nm bin is highest for mean particle sizes below approximately 30 nm. The bias in the 184.6 nm bin is mostly strongly negative, but with an increasing trend for increasing mean particle sizes. The bias in the size bins <100 nm show no strong dependence on the geometric standard deviation for

geometric standard deviations above approximately 1.8. In contrast, the bias of the concentrations particularly in the 113.5 nm and 184.6 nm size bin show rather strong dependence on the geometric standard deviation. A clear dependence of the concentration in the 300 nm bin on the geometric standard deviation could not be observed. The concentrations, measured in the largest size bins may also have been affected by the presence of larger, multiply charged particles. However, during the measurement period, the number concentration of particles >300 nm only amounted to on average 0.9% of the total number

concentration (result from MPSS).

The bias analysis shows that the uncertainty of the number size distributions measured with the Partector Pro is dependent on the size distribution. The most accurate results can be expected for particle size bins with midpoints of 113.5 nm and smaller, when the mean particle size is larger than 30 nm and the geometric standard deviation is larger than 1.8. During the measurements, presented here, 58.5% of the hourly averages fulfilled these criteria. Number concentrations in the size bins

with midpoints at 184.6 nm and 300 nm should be considered as an indicator of the size distribution rather than an accurate measurement.

## 4.2 Number concentration

The Partector Pro delivers a value for the total number concentration directly. Alternatively, it can be calculated by integrating the number size distribution ($dlog(d_p)$ = 0.211), which yields the same results. In order to compare the total number

concentrations measured with the Partector Pro and the MPSS, the size distribution data, corrected for the inlet losses (see





Figure 2) were integrated for both devices. Equation (1) was used for the correction of the Partector Pro data. For the MPSS, only the concentrations of particle sizes up to 300 nm were integrated to ensure that the considered sizes match the nominal size range of the Partector Pro. Since the number concentration of particles >300 nm only amounted to 0.9% of the total concentration, this size limitation of the MPSS data only had a minor impact. CEN/TS 16976:2016 and prEN 16976:2023

prescribe that a CPC for the measurement of ambient number concentrations shall be calibrated in a laboratory under well controlled conditions with a silver test aerosol of different concentration levels and with either a calibrated CPC or an aerosol electrometer as reference. The performance of the test CPC is evaluated by a linear fit of its data and the reference data. The linear fit shall be forced to go through the origin and the resulting slope shall be within $1 \pm 0.05$. Although the test aerosol is different and the integrated number concentration of an MPSS is used as reference here, the same fit procedure and the criteria

for the slope were applied to the hourly averaged concentration data measured with the MPSS and the Partector Pro during the entire measurement period. Figure 6 shows a scatter plot, in which the hourly averages, measured with the Partector Pro are plotted against the MPSS reference data along with two linear fits. On the one hand, all data were used for the fit and on the other hand, only concentrations below 60,000 1/cm³. During the entire measurement period, there were three hourly average concentration values above 60,000 1/cm³, for which the Partector Pro results deviated noticeably from the number

concentrations measured with the MPSS. If these data points are excluded from the fit, the slope is 0.998 and the regression coefficient $R^2 = 0.996$. Consequently, the number concentrations, measured with the Partector Pro were in excellent agreement with the MPSS data. When all data were used for the fit, the slope is reduced to 0.977 and the regression coefficient to $R^2 = 0.989$.

**Figure 6**

During the measurements presented here, the Partector Pro thus fulfilled the requirements as set in CEN/TS 16976:2016 and prEN 16976:2023, even though the measurements were not conducted under well controlled laboratory conditions and not with a well-defined test aerosol as requested by the Technical Specification/standard, but under field conditions and with

atmospheric aerosol. This high level of agreement between number concentrations measured with Partector Pro and MPSS is astonishing, given the fact that the manufacturer specifies the uncertainty range for the number concentration measurement as ±30%. It is possible that this excellent agreement may not hold at other measurement sites and/or other applications. Nonetheless, the very high linearity of the data Figure 6 are proof of the good performance of the instrument.

**4.3 Geometric mean particle diameter**

The Partector Pro provides the geometric mean particle size based on the measured size distribution. Figure 7 shows a scatter plot of the hourly average mean diameter values, delivered by the Partector Pro versus the geometric mean diameter values measured with the MPSS. It can be seen that the agreement is best for mean diameters between 20 nm and 50 nm with deviation



mostly within ±15%. For larger particles, a higher scatter can be observed. For mean particle diameters approximately >70 nm, the Partector Pro tends to overestimate the particle size. Size distributions with mean particle sizes approximately >70 nm were

nearly all bimodal, with the first mode in a size range of 20-30 nm and a second, higher mode around 100 nm. It is likely that the data inversion cannot properly account for bimodal size distributions.

**Figure 7**

Figure S 5 in the supplementary material plots the bias of the mean particle size determined with the Partector Pro compared with the MPSS as a function of the geometric standard deviation of the aerosol. The graph shows a moderate but increasing trend of the bias with increasing geometric standard deviation, i.e. the mean particle size is underestimated for narrower distributions and overestimated for wider distributions. On average, the best agreement was found for geometric standard deviations between 1.8 and 2.6 with random deviations mostly within ±10% with a few outliers. For geometric standard

deviations below 1.8, the mean diameters were always underestimated by up to 12% by the Partector Pro with a clear decreasing trend with decreasing standard deviation. For geometric standard deviations larger than 2.6, the mean diameters were almost always overestimated by up to 32% and increasing with increasing geometric standard deviations. It should, however, be noted that such low or high standard deviations are rather rarely encountered in atmospheric aerosols. During the measurements conducted here, only 9.0% of the hourly average number size distributions showed geometric standard

deviations larger than 2.6 and 7.4% smaller than 1.8, i.e. the majority of 83.6% of the size distributions were such that the mean diameter could be estimated by the Partector Pro with uncertainties mostly within ±10%.

**5 Summary and conclusions**

The number size distributions, total number concentrations and mean particle sizes of atmospheric airborne particles were measured in parallel with a Partector Pro (size range 10-300 nm in 8 size bins) and an MPSS (size range 10 – 800 nm in 122

size bins) at an urban background site in Mülheim-Styrum in Germany for a period of 70 days. All size distributions were corrected for losses in the inlet system and hourly mean distributions calculated. The data evaluation showed that the uncertainty of the size distributions measured with the Partector Pro is dependent on the mean particle size and geometric standard deviation of the aerosol. The best agreement was observed for the particle size bins between 16.3 nm and 113.5 nm, for which the bias per size bin was mostly well within ±25%. Larger discrepancies were only observed during rather rare cases

when the geometric standard deviation was below approximately 1.8. On average, the bias of the daily averages was -8.6%, -6.4%, -6.3%, -8.3% and +13.1% for the 16.3 nm, 26.4 nm, 43.0 nm, 69.8 nm and 113.5 nm size bins, respectively. The bias in the 10 nm size bin was -43.4%. As a result, the number size distributions measured with the Partector Pro, averaged over the entire measurement period, fulfilled the criteria in CEN/TS 17343:2020 for particle sizes below <100 nm. However, the daily individual values sometimes exceeded the limits. The specified max. deviations are ±50% for sizes below 20 nm and ±10%



for sizes between 20 nm and 200 nm. The bias for the concentrations in the size bins with midpoints of 184.6 nm and 300 nm were larger, i.e. -57.7% and -23.6%, respectively. The deviations in the 184.6 nm bin were much more strongly dependent on the mean particle size and the geometric standard deviation of the aerosol. However, all the observed dependencies of the bias on the mean particle size and the geometric standard deviation did not follow simple relationships and were different for each size bin.

Hourly averaged total number concentrations, measured with the Partector Pro were in excellent agreement with the number concentrations measured with the MPSS (slope: 0.998, $R^2 = 0.996$), when only number concentration values below 60,000 1/cm³ were considered. When higher concentrations were included in the linear fit, the agreement was slightly deteriorated (slope: 0.977, $R^2 = 0.990$). However, both results are well within the limits, defined in CEN/TS 16976:2016 and prEN 16976:2023 for the performance of CPCs for atmospheric measurements.

The hourly averages of the mean particle diameter, determined by the Partector Pro were in good agreement with the geometric mean particle diameters, determined with the MPSS (slope: 1.066, $R^2 = 0.948$). The agreement was moderately dependent on the particle size distribution. For mean particle sizes between 20 nm and 50 nm, the agreement was mostly within ±15%, whereas the mean particle size was clearly overestimated, when the mean particle size exceeded 70 nm. The agreement was additionally dependent on the geometric standard deviation of the test aerosol. Best results were obtained for the most abundant

range of $1.8 \leq \sigma_g \leq 2.6$. Geometric standard deviations below 1.8 led to an underestimation and geometric standard deviations larger than 2.6 to an overestimation of the mean particle size.

In summary, the Partector Pro accurately measured the number concentrations of the atmospheric aerosol at the measurement site in Mülheim-Styrum during the measurement period. Additionally, it provided very good estimates of the mean particle size. It remains to be investigated, whether the good agreement with the MPSS results is representative or specific to this

measurement campaign and location. The Partector Pro further provided reasonably good estimates of the number size distribution of particles smaller than 100 nm, but relatively large and systematic discrepancies for particles in the two largest size bins. The device also measures the lung deposited surface area concentration, but these results have not yet been further analysed. Although the Partector Pro cannot replace an MPSS system concerning the measurement uncertainty and size resolution, it is a very useful tool for many applications, where the use of an MPSS is not feasible due to its size and/or price,

e.g. mobile measurements or in locations with limited space. Based on the results of this study, it can be concluded that the Partector Pro may also be an alternative to measurements of the total atmospheric number concentration with a CPC, as it does not require frequent maintenance or a refill of the working fluid reservoir. In such a case, the Partector Pro would not only be cheaper, but also provide additional information on the size distribution, mean particle size and lung deposited surface area concentration. However, potential effects of not operating the Partector Pro in an air conditioned container and without sample

conditioning are yet to be investigated. It should additionally be noted that this is the first study on the performance of the first generation of the Partector Pro and information on the long-term stability over a period extending 70 days is yet lacking.



**Acknowledgements**

The authors would like to thank Prof. Dr. Martin Fierz (University of Applied Science North Western Switzerland and naneos GmbH) for very helpful input on the operating principle of the Partector Pro and the data analysis. The authors would further

like to thank Dr. Sebastian Schmitt and Dr. Amine Koched of TSI for their technical support.

The MPSS was funded by the German Federal Ministry for Economic Affairs and Climate Action (BMWK) based on a decision by the German Bundestag. Performance of the measurements and data evaluation have received funding from the Landesamt für Natur, Umwelt und Verbraucherschutz (LANUV) of the federal state of North Rhine Westphalia (NRW). The support by BMWK and LANUV is gratefully acknowledged.

**Author contributions**

CA, AMT and HK conceived the study. CA and HK set up the instruments at the measurement site and were responsible for the performance of the measurements. CA was responsible for the data analysis and drafting of the manuscript. HK and AMT were responsible for maintenance and calibration of the MPSS and Partector Pro. AMT supported data analysis and drafting of the paper.

**Competing interests**

The authors declare no conflict of interest.

**Data availiability**

All Partector Pro and MPSS raw data as well as the hourly and daily mean values are available at Zenodo:
https://doi.org/10.5281/zenodo.8234790

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



## Figures

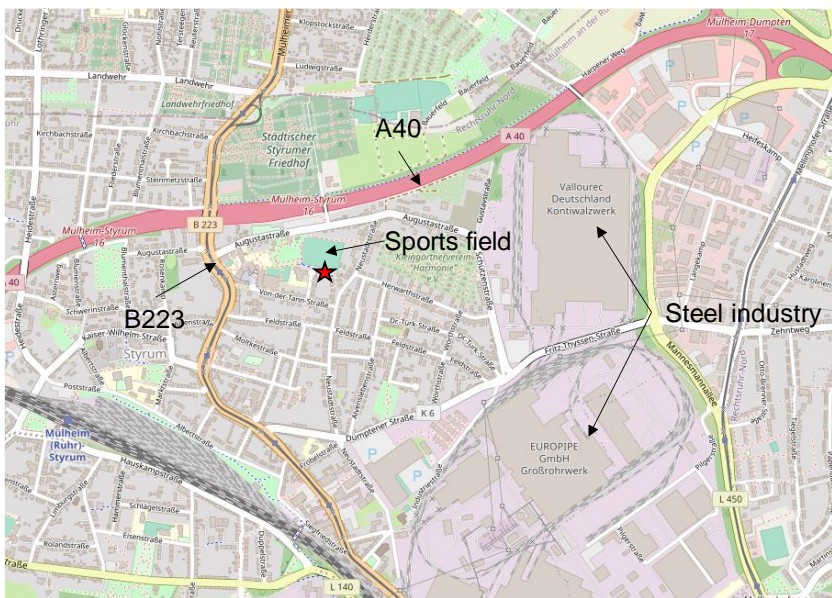

**Figure 1: Location (red asterix) and direct vicinity of the measurement location in Mülheim-Styrum (map © OpenStreetMap contributors 2023. Distributed under the Open Data Commons Open Database License (ODbL) v1.0.)**



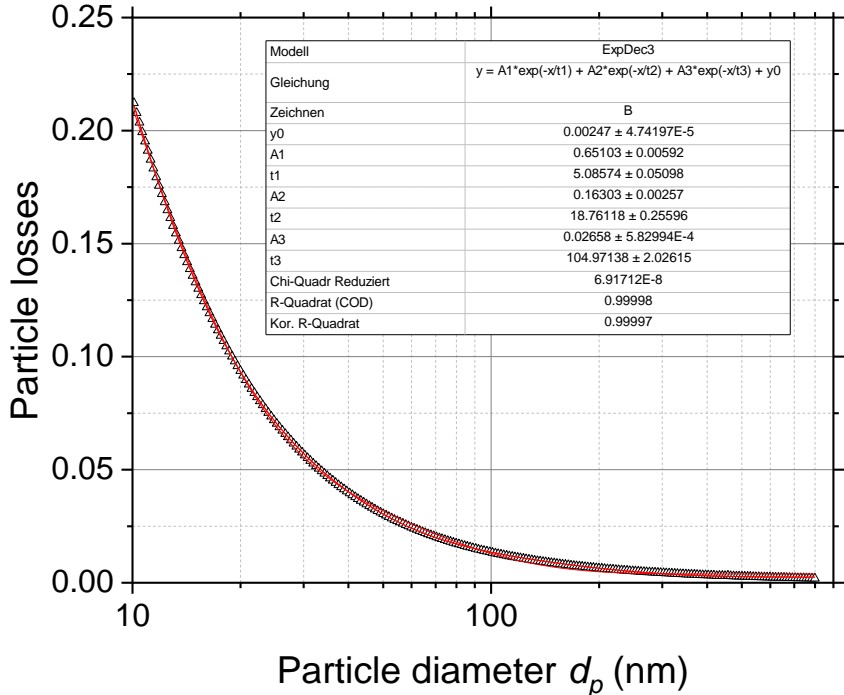

**Figure 2: Particle losses in the sampling system as a function of particle size and exponential fit**





**Figure 3: Contour plots of the particle number size distributions (hourly averages) measured with the MPSS and Partector Pro during the measurement period**



**Figure 4: Mean particle size distribution during the measurement period, measured with the MPSS and Partector Pro; grey shaded area illustrates the allowed uncertainty range according to CEN/TS 17343**





Figure 5: Bias of the number concentration per Partector Pro size bin as hourly (black circle) and daily (red triangle) averages; error bars display the standard deviations





**Figure 6: Scatter plot of the number concentrations (hourly averages) measured with the Partector Pro vs. MPSS with linear fit for all data (red) and for concentrations below 6.0*10⁴ 1/cm³**

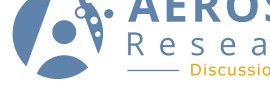

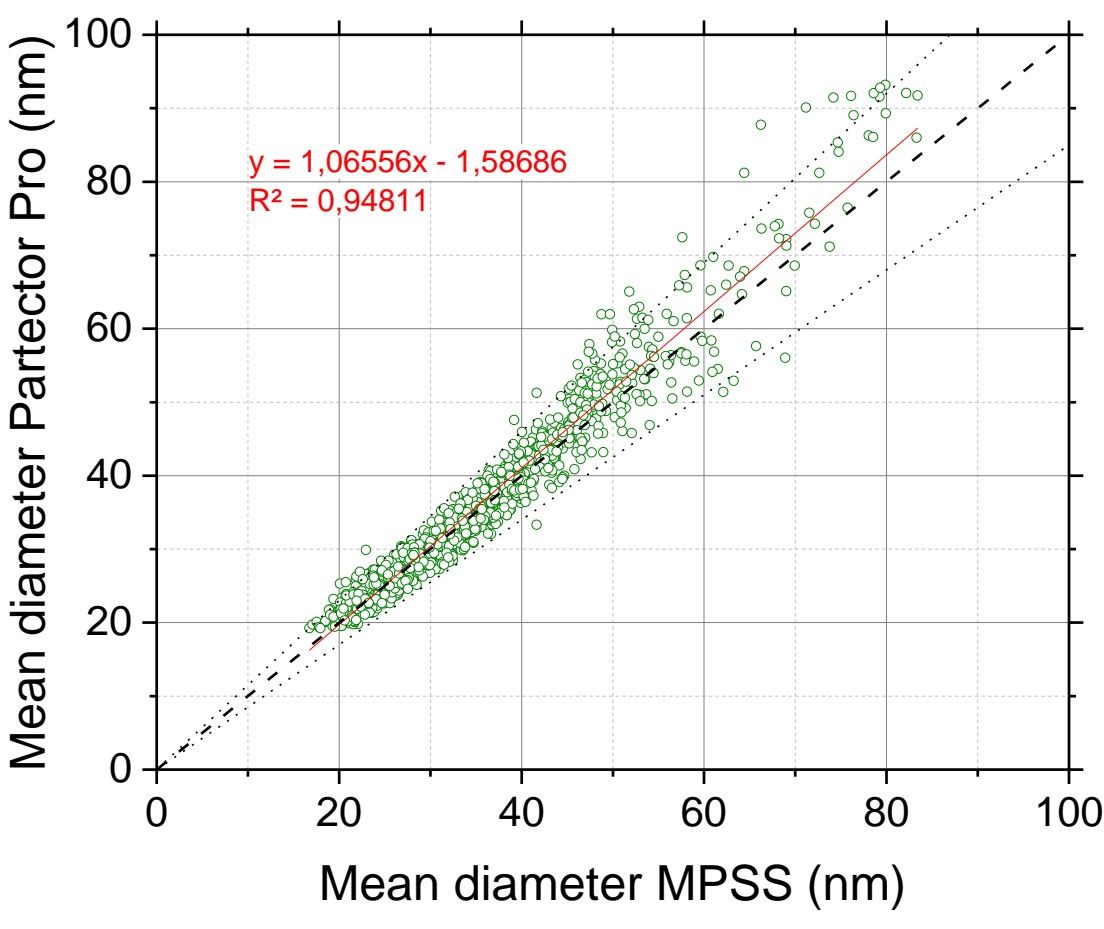

**Figure 7: Scatter plot of the geometric mean particle size (hourly averages) measured with the Partector Pro vs. MPSS with linear fit (red line); 1:1 agreement is indicated by the dashed line, dotted lines mark the limits of ±15% deviation**



**Table 1: Midpoints of the Partector Pro size bins with their lower and upper limit and corresponding MPSS size bins, used for summarizing and determining the bias**

| Partector Pro | | | MPSS | |
|---|---|---|---|---|
| **Midpoint (nm)** | Lower limit (nm) | Upper limit (nm) | Smallest bin (nm) | Largest bin (nm) |
| **10.0** | 7.8 | 12.7 | 10.18 | 12.63 |
| **16.3** | 12.7 | 20.7 | 13.1 | 20.2 |
| **26.4** | 20.7 | 33.7 | 20.9 | 33.4 |
| **43.0** | 33.7 | 54.8 | 34.6 | 53.3 |
| **69.8** | 54.8 | 89.0 | 55.2 | 88.2 |
| **113.5** | 89.0 | 144.7 | 91.4 | 140.8 |
| **188.6** | 144.7 | 235.2 | 145.9 | 232.9 |
| **300.0** | 235.2 | 382.4 | 241.4 | 371.8 |