# Peer review of "Evaluation of a Partector Pro for atmospheric particle number size distribution and number concentration measurements at an urban background site"

_Aerosol Research, 2023_

## Author Response (AR1)

We would like to take the opportunity to thank both reviewers for their review and thoughtful comments. They were very helpful to further improve the manuscripts. Our responses to the reviewers' comments can be found below.

Reviewer 1:

The MS compares an SMPS system with a Partector Pro monitoring instrument. It is based on contemporary experimental and evaluation methods. The results and conclusion could represent valuable contributions to the growing research field on UFP, and the MS could be worth publishing after major revision. This can be hopefully done by a rigorous and substantial complementation on the present version.

Thank you for your careful review of the paper. We addressed all your comments in the manuscript and below.

Major concerns

1a The Introduction correctly highlights the role of UFP in health effects of PM. Despite, the comparison of atmospheric concentrations is performed only for particle number concentrations in the 10–300 nm diameter range. I suggest that additional separate comparisons, at least of the concentrations for d<100 nm are included as well.

Thank you for pointing this out. The main intention of the paper is to evaluate the performance of the Partector Pro against an MPSS and thus only considered the complete overlapping size range for the comparison of the number concentration. However, we agree with the reviewer that with the ongoing discussion on potential health effects of UFP, an additional comparison of the UFP-concentration is useful. A second scatter plot was thus added to Figure6 that covers only the UFP concentrations. The comparability for this size range is nearly equally good as for the total size range. A brief discussion on this was added to the text on page 10.

1b Similarly, the comparison is performed only on 1-h averaged data. It would be informative in a research journal (on top of the international technical specifications) to supply the time series of bias or at least the average bias for the corresponding individual data (of those with close time stamps) as well.

Plotting the bias data with higher time resolution leads to the same conclusions, but the plots become more scattered and difficult to read. Nonetheless, we agree that plotting the time series of the bias for the individual data can help gaining additional insight. Time series of the bias in the individual size bins, for the mean diameter and the total concentration were thus added as new Figure S3 to the supplemental material.

1c Furthermore, it is not defined what the authors exactly mean on the expression bias.

The definition of the bias has been added as equation (2) on page 8.

If it is related to concentration ratios then it should be discussed that the bias in % depends on the absolute values of the related concentrations. The latter is mainly governed by the size distribution, and might be related "to higher, systematic differences for larger particles" (as given in the Abstract).

We believe that this is resolved with the addition of the equation for the bias. The higher systematic differences for larger particles are apparent from Figure 5.

2. Figure 4 shows the mean size distributions obtained by both methods over 70 days. They contain a large Aitken mode and a smaller accumulation mode. This follows from the type of the sampling site (urban background). Nevertheless, the shape of the individual distributions could be different from the mean for some shorted time intervals, e.g. for nights. It is not exactly clarified in the MS how the geometric mean diameter (GMD) was determined. Since the authors seem to present just one GMD value for one size distribution, the GMD could be related to the whole particle population. This could be misleading, since the relative increase of the accumulation mode (e.g. during nights) could apparently shift the GMDs to larger values. Vice versa, it could be related to "Best results were found for the most abundant size distributions with geometric mean particle diameters ≥30 nm…" (in the Abstract). The authors are requested to clarify this, to discuss this possible effect and to amend the MS by separate comparisons for daylight times and nights if necessary.

The geometric mean diameter is one of the parameters that is directly delivered by the Partector Pro for the entire width of the measured size distribution. Consequently, this value was compared with the geometric mean diameter from the MPSS in the same size range. A further distinction of the different modes of the distribution with the Partecotr Pro is not very meaningful, due to its rather low size resolution. A short paragraph was added to describe how the GMDs were obtained.

Size distributions during day and night were calculated and show the behavior described by the reviewer. The differences in the two modes were, however, not very large. Consequently, we decided to not show them in the manuscript, since the focus of the manuscript is on the comparison of the two instruments and not so much a characterization of the aerosol at this specific location. For this purpose, we don't feel that separate plots for daytime and nighttime add much to the discussion. The key message for the geometric mean diameter can be seen from Figure 7, i.e. the mean geometric mean diameter, determined by the Partector represents the geometric mean diameter, determined with the MPSS well for mean diameters below approximately 50 nm, but is overestimated for larger particles.

A short paragraph on the two modes and the changes during day and night was added.

3. The title should include: …particle number size distribution and concentration measurements… or something similar. (Likewise, in the Abstract: Particle number size distributions…)

Thank you for this comment, the title and the abstract have been changed accordingly.

Minor comments

4. Abstract: The authors may want to replace "below 113.5 nm" by "between 10 and 113.5 nm" since many PMSSs have their lower measured diameter <10 nm.

Good point, we changed the manuscript accordingly.

5. L37: "can also arise naturally from nucleation processes" should be clarified with respect to the fact that it is not sensible to separate secondary aerosol particles (e.g. from nucleation process) into natural and anthropogenic sources.

The following sentence was added to clarify this:

Although nucleation occurs in the atmosphere as a natural process, the precursors that lead to nucleation can still be of both anthropogenic or natural origin.

6. L85: Provide the height above mean sea level.

The height above sea level (40 m) was added on page 3.

7. Figure 2: the legend should be also in English.

Thank you for pointing this out. We now also translated the legend.

8. L222 and 223: Is it +/-50% and +/-10%?

Correct, we added "±" where appropriate.

9. L293: Consider writing 1.00+/-0.05.

Changed as suggested by the reviewer

10. It is advised that the scatter plots (Figs. 4 and 5) have squared layout. See also comment no. 13.

Changed as suggested by the reviewer

11. Revisit you rounding off strategy; some examples are in L253: -57.7% and Table 1: 371.8 nm.

The values provided in Table 1 are those provided by the manufacturers and were thus not change. Otherwise, we rounded all values to one significant digit.

12. Figure 6: Resolve the meaning of the dashed line.

Thanks for pointing this out. The dashed line refers to the range of ±30% agreement. This information was added to the figure caption.

13. The MS misses a basic reference of Wiedensohler et al., (2018): Mobility particle size spectrometers: Calibration procedures and measurement uncertainties, Aerosol Science and Technology, 52:2, 146-164, DOI: 10.1080/02786826.2017.1387229.

Thanks for the hint. The reference was now added at several points in the text.

Reviewer 2:

The paper presents an excellent comparison between state-of-the-art instrumentation and an innovative system for ultrafine particulate sampling. This new instrumentation, transportable, with low energy impact and easyr operation (e.g. costs, management) can provide an important contribution to the monitoring of UFP. The monitoring of fine particulate matter has become of primary importance for the scientific community but also due to the importance it has achieved for changes to EU legislation. In this paper, the results presented by state-of-the-art instrumentation and this new system are analytically compared, providing encouraging results for monitoring with compact and easier to manage systems. The paper presents statistics of observations over a period greater than two months in an extra-urban area thus providing solid results to characterize both temporal and concentration variability (also note the analysis of the spike event).

Thank you for your kind words and the thoughtful review. Your comments are addressed in the manuscript and below.

From my point of view the work can be considered for publication by encouraging the authors to make the figures clearer by following the following notes:

Fig1: redo the figure in table format where the geography of the station is also shown (E. row1 col1 North Europe, row1 col2 NW Germany, row 2 as actual fig1 but better resolution)

We changed the figure as suggested by the reviewer, so that readers who are not familiar with the Ruhr area/Western Germany can now better identify the location of the measurement location.

Fig3: redo the figure with a shorter color scale (e.g. max 1x10^5) otherwise the figure is useless

Thank you for this suggestion, we changed the figure accordingly.

Fig4: cut Xaxis at 300nm

We changed the figure as suggested.

Fig5: change the Yaxis from -100 nm to 100 nm

We changed axis to a range from -100% to +100% (not nm)

Fig6: add in the caption (below 6.0*10^4 (1/cm3) DASHED LINE

The dashed line refers to the range of ±30% agreement. We added this information in the figure caption.

With these considerations I believe that the paper can be published.

Martin Fierz:

Disclaimer: I am one of the founders of naneos particle solutions, and I have developed the measurement + data inversion algorithm of the Partector 2 pro.

This paper presents a nice & clear comparison of an SMPS and Partector 2 Pro.

For us as instrument developers, it was very valuable, as the data presented here was shared with us, and we used it to further improve the data inversion algorithm of the Partector 2 Pro. With the latest firmware, the underestimation in the 10nm channel shown in figures 4+5 would be less severe, but would still exist (it would be about half as big as reported here). Further refinements to the calibration procedure might help to improve the data inversion further, but this is still work in progress. In any case, having large datasets as the one presented here is always important/necessary for continuous improvement, and we are grateful to the authors for performing this study and sharing its results with us.

Martin Fierz

Thank you, Martin. I added a sentence at the end of the conclusions concerning the improved algorithm and firmware.